# The associations between autistic traits and disordered eating/drive for muscularity are independent of anxiety and depression in females but not males

**John Galvin**[1], **Elizabeth H. Evans**[2], **Catherine V. Talbot**[3]*, **Claire Wilson**[4], **Gareth Richards**[5]

1 Department of Psychology, Birmingham City University, Birmingham, United Kingdom, 2 Department of Psychology, Durham University, Durham, United Kingdom, 3 Department of Psychology, Bournemouth University, Poole, United Kingdom, 4 School of Education, University of Strathclyde, Glasgow, United Kingdom, 5 Faculty of Medical Sciences, School of Psychology, Newcastle University, Newcastle upon Tyne, United Kingdom

* ctalbot@bournemouth.ac.uk

**Data Availability Statement:** The data underlying the results presented in the study are available from the OSF website for the project (osf.io/kz6x4).

## Abstract

Previous research has shown a positive correlation between autistic traits and eating disorder symptoms, and this relationship appears to be independent of co-occurring mental health status. The current study followed a pre-registered analysis plan with the aim to investigate a previously unconsidered factor in the relationship between autistic traits and disorders of eating and body image: the drive for muscularity. Participants (N = 1068) completed the Autism Spectrum Quotient (AQ), Hospital Anxiety and Depression Scale (HADS), Eating Attitudes Test-26 (EAT-26) and Drive for Muscularity Scale (DMS). Positive correlations between AQ and EAT-26 and AQ and DMS were observed. In females, AQ remained significantly correlated with EAT-26 and DMS when controlling for co-occurring anxiety and depression symptoms, but this was not the case in males. These findings demonstrate the moderating role of sex, and the need to consider autistic traits in individuals diagnosed with, or at a heightened risk for, disorders of eating and body image.

## Introduction

Autism Spectrum Conditions (hereafter autism) are neurodevelopmental conditions characterised by restrictive/repetitive behaviours and difficulties with social communication and interaction. The core characteristics of autism vary considerably between individuals in terms of severity and presentation, and co-occurring conditions such as anxiety, depression, and eating disorders (EDs) are common [1,2]. Prevalence of autism in the UK is estimated to be at least 1%, with approximately four males receiving a diagnosis per one female [3]. Higher prevalence in males could reflect a distinct phenotypic difference between the sexes [4], a systematic bias towards the typical presentation, measurement, and diagnosis of autism [5], or a combination of these factors. In addition to clinically diagnosed autism, traits associated with the condition are also commonly investigated in non-clinical populations. In these types of

**Funding:** Funding was provided by the School of Psychology, Newcastle University for participant recruitment on Prolific. The funds were spent from allocated funding for staff projects (Elizabeth H. Evans research account). The funder had no role in the study design, data collection and analysis, decision to publish, or preparation of the manuscript.

**Competing interests:** The authors have declared that no competing interests exist.

studies, autism is typically conceptualised as a dimensional condition representing one end of a continuum. Consistent with this, research shows that autistic traits are continuously distributed in the general population [6] and that autistic people tend to score higher than non-autistic people on these measures [6,7]. In line with autism diagnoses, most general population studies show males score higher on average than females on the AQ [6].

## Autistic traits and disordered eating

It has been noted for several decades that autistic individuals, particularly autistic women, have an elevated risk of EDs such as anorexia nervosa (AN) [8–10] and that individuals diagnosed with both autism and AN describe these conditions as being deeply interlinked [11]. Furthermore, autism shares certain clinical features with eating disorders, such as cognitive inflexibility, sensory sensitivities, restrictive and repetitive behaviours, and impaired social cognition and emotion regulation [12,13]. These similarities extend beyond clinical groups, with small-to-moderate sized positive correlations between autistic traits and disordered eating symptoms found in general population studies [14]. Considering that males are more likely to receive an autism diagnosis whereas females are more likely to be diagnosed with EDs, there has been recent interest in the role that biological sex may play regarding the association between autistic traits and disordered eating symptoms [15,16]. Barnett et al. [15] reported that the correlation between self-reported autistic traits and disordered eating symptoms is significantly stronger in general population females than general population males, and that the observed relationship remained statistically significant (in the combined male + female sample) after controlling for concurrent symptoms of anxiety and depression.

## Muscle dysmorphia

Although not included in previous versions of the Diagnostic and Statistical Manual of Mental Disorders (DSM), muscle dysmorphia (MD) is now a recognised condition in the DSM-5 [17], and considered a specifier of Body Dysmorphic Disorder (BDD). According to the DSM-5 definition, BDD is characterised by an extensive preoccupation with perceived flaws in appearance. MD is diagnosed when an individual's negative evaluative body image and appearance modification cognitions and behaviours are focused on a perceived need to become bigger and more muscular (i.e., a "drive for muscularity"), regardless of their current size. Research has noted similarities between the clinical features of MD and eating disorders: individuals diagnosed with both categories of disorder typically show rigid dietary restraint and restriction, excessive, regimented physical activity, the habitual subjugation of biological cues to hunger and satiety, cognitive distortions of body size and shape perception, and high levels of body dissatisfaction [18,19].

Moreover, a growing body of literature [20,21] proposes that the maintenance of MD symptoms can be best understood through the lens of the transdiagnostic model of eating disorders [22]. This model is based on the premise that a wide range of disordered eating symptoms and syndromes stem from one unified network of variables, i.e., a single core psychopathology (comprising over-evaluation of eating, weight, and shape, and their control, and dietary restraint). BDD, and, by extension MD, tends to be underdiagnosed [23], and these syndromes are commonly misdiagnosed as other disorders, such as anxiety or depression [24]. Consequently, the conditions are often left untreated or inadequately treated. Furthermore, misdiagnosis may also occur because MD focuses on increased body size, not body size reduction (as is the case in restrictive eating disorders such as AN). The disordered behaviours commonly associated with MD could therefore be more likely to be perceived as requiring less immediate medical attention.

## Body image disturbance and drive for muscularity

Body image can be defined as perceptions, thoughts, and feelings about one's body [25]. The literature on body image disturbance has historically focused on female body ideals, such as a drive for thinness, as opposed to male body ideals, such as a drive for muscularity [26,27]. The drive for muscularity concept was first proposed by McCreary and Sasse [26] to reflect male body image concerns and as a parallel to the already established drive for thinness in females. Several studies have shown that drive for muscularity (often measured with the Drive for Muscularity Scale [DMS] [26]) is related to a range of negative outcomes such as poor emotion regulation [28], low self-esteem [29], heightened risk of depression [30], eating disorders [31], and muscle dysmorphia [32]. In the context of bodybuilding, drive for muscularity is associated with significant dietary restraint, with sugar and caloric intake being reduced and protein intake being increased [33,34]. This eating pattern can lead to binge eating behaviours [35] and excessive consumption of potentially dangerous supplements such as creatine and amino acids [27]. Just as disordered eating symptoms are distributed dimensionally throughout the general population and act as indices of clinically-relevant phenomena in aetiological studies, drive for muscularity–an index for the size- and body-focused cognitions and behaviours that typify MD–is also distributed dimensionally in both male and female general population samples [36].

In relation to sex differences in drive for muscularity, one study found that 22% of men and 5% of women aged 18–24 reported muscularity-oriented disordered eating [37]. However, other recent studies [e.g., 38] suggest that the prevalence of muscularity pursuit could also be on the rise in females. Societal pressures in Western cultures have shifted women's body ideals towards a "toned" appearance, characterised by both thinness and muscularity, a physique that is increasingly idealised through social media content [39]. For example, the trend known as 'fitspiration' (fitness and inspiration) promotes the idea that there is a certain "look" to being fit and healthy (i.e., a thin and toned body) [40,41], and research shows that posting fitspiration images on social media is directly correlated with maladaptive eating and exercise behaviours in women [42].

## Rationale for the current investigation

There are several reasons to suspect that autistic traits may be positively associated with a drive for muscularity in general population males and females. Individuals with high autistic traits are more likely to report 'black-and-white' thinking (i.e., polarised or inflexible thinking patterns) and intense interests around their food, weight, diet, and exercise [12]. They are also more likely to have rigid/routinised behaviours and experience specific sensitivities to their bodily changes and emotions, which may impact physical training or consumption decisions [43,44]. Like people with clinical diagnoses of AN or MD, individuals with high levels of autistic traits demonstrate a tendency to focus on local details at the expense of global processing [45], a phenomenon otherwise known as "weak central coherence" [46]. High autistic traits are also correlated with increased difficulties with interoceptive sense [47] and weak integration of visuotactile-proprioceptive information about the body [48]. Furthermore, it is well established that both autistic traits and body image concerns are linked to specific patterns of executive functioning and the ability to interpret judgements [49]. For instance, research shows that females with AN do not appear to show altered body representation in general, but rather a top-down cognitive-affective distortion in evaluating their own body [50]. These and other results [12,13,16] provide evidence that the unique cognitive profiles exhibited by individuals with high levels of autistic traits may increase the risk of psychopathology associated with unhealthy body image and maladaptive eating and bodily behaviours.

Given the previously discussed literature showing a higher prevalence of autism in males and ED psychopathology in females, debates continue as to whether the higher rates of autistic symptoms observed in ED populations are truly due to underlying autism or may instead be explained by shared psychopathology across the two conditions [13]. For instance, high levels of anxiety and depression could exacerbate social and flexibility difficulties in ED populations, which could, in turn, be mistaken for autistic symptomology. When modelling the associations between autistic traits and disordered eating/drive for muscularity, it is therefore important to account for other mental health symptoms that may otherwise explain these relationships. Although Barnett et al. [15] reported that the correlation between autistic traits and disordered eating symptoms remained statistically significant after controlling for co-occurring anxiety and depression, it remains to be seen whether this is also the case regarding drive for muscularity.

Considering the already established sex differences in autistic traits (male > female), disordered eating (female > male) and drive for muscularity (male > female), as well as the positive correlations observed between autistic traits and disordered eating in both males and females [15], the current study aimed to replicate and extend previous work in this area. We pre-registered our hypotheses and analysis plan on the Open Science Framework (osf.io/kz6x4) and aimed to replicate the main findings of Barnett et al. [15] regarding the positive correlation between autistic traits and disordered eating symptoms. We also aimed to extend this research by investigating the association between autistic traits and drive for muscularity. To our knowledge this is the first study to consider the relationship between autistic traits and drive for muscularity. Our main hypotheses were that 1) autistic traits would be positively correlated with disordered eating symptoms and drive for muscularity, and 2) the correlations would remain statistically significant after controlling for concurrent symptoms of anxiety and depression.

## Method

Ethical approval was provided by the Faculty of Medical Sciences Research Ethics Committee, Newcastle University (approval number: 7601/2020).

### Design

The study employed a correlational design. The predictor variable was autistic traits, and the outcome variables were disordered eating symptoms and drive for muscularity.

### Sampling and procedure

An *a priori* power analysis with 80% power and alpha set at $p < 0.05$ was conducted using G*Power 3.1 [51]. Based on the effect sizes reported by Barnett et al. [15] for the correlation between AQ (total score) and EAT-26 (total score) in a non-clinical population, the required sample sizes determined were $n = 634$ for males and $n = 73$ for females. However, some analyses (i.e., those for which covariates were included) would require more participants, and we had no prior indication of the strength of the association between autistic traits and drive for muscularity. We therefore aimed to obtain a larger overall sample size.

An online survey hosted on Qualtrics was advertised on student participation panels at the authors' institutions, as well as on social media. Participants were presented with the information about the research and were required to provide informed consent before proceeding with the study. Participants were renumerated with course credit (student sample) or the opportunity to enter a prize-draw for a £25 Amazon voucher (social media sample), with one Amazon voucher being drawn for every 50 participants recruited.

Seven hundred and twenty-two people accessed the survey, 537 of whom completed at least one of the questionnaire measures ($n$ = 113 males, $n$ = 415 females, $n$ = 2 preferred not to say, and $n$ = 7 preferred to self-describe). As we had not achieved the target sample size for males, we collected data from a further $n$ = 558 males from the participant recruitment website Prolific (www.prolific.co). Prolific was chosen because recent studies suggest the platform produces better quality data than other participant recruitment panels [52,53]. However, as it is known that there can still be some low-quality responses when using crowdsourcing platforms [54], we included two additional items, randomly placed within the questionnaires, to act as attention checks. Eighteen participants failed two out of two attention checks and were subsequently removed. This resulted in a sample size of $n$ = 540 males recruited from Prolific.

The addition of the Prolific data to the student and social media data resulted in a sample size of $N$ = 1077 (males $n$ = 653, females $n$ = 415, prefer not to say $n$ = 2, prefer to self-describe $n$ = 7). However, with the focus on sex differences, the latter nine participants were not included in subsequent analysis. The final sample size was therefore $N$ = 1068 (males $n$ = 653, females $n$ = 415), with an age range of 18–75 years ($M$ = 28.65, $SD$ = 11.96). Most participants described themselves as being of White ethnicity ($n$ = 818, 76.6%). The remaining participants described themselves as Asian/Asian British ($n$ = 85, 8%), Black/Black British ($n$ = 50, 4.7%), Black Other ($n$ = 40, 3.7%), Hispanic or Latino ($n$ = 32, 3.0%), Mixed Ethnicity ($n$ = 19, 1.8%), Chinese ($n$ = 13, 1.2%), or Middle / Near Eastern ($n$ = 10, 0.9%). Five hundred and sixty participants (52%) were students and 508 (48%) were not students.

Participants were asked to report whether they had been diagnosed with any of the following conditions: autism (diagnosed $n$ = 40, 3.7%; suspected $n$ = 95, 8.9%), anxiety (diagnosed $n$ = 236, 22.1%; suspected $n$ = 336, 31.5%), depression (diagnosed $n$ = 198, 18.5%; suspected $n$ = 256, 24.0%), an eating disorder (e.g., Anorexia Nervosa, Binge Eating Disorder, Bulimia Nervosa) (diagnosed $n$ = 43, 4.0%; suspected $n$ = 123, 11.5%) and muscle dysmorphia (diagnosed $n$ = 4, 0.4%; suspected $n$ = 43, 4.0%). Missing data ranged from 0.2% (over evaluation of weight and shape, $n$ = 2 missing) to 1% (AQ total, $n$ = 11 missing). BMI ranged between 12.17 and 84.87 ($M$ = 24.91, $SD$ = 6.82), although five implausible values (BMI ranged between = 8.77 and 11.43) were removed (please note that these participants also did not report a diagnosed or suspected eating disorder, suggesting data entry errors).

## Materials

Participants reported their sex (male, female, prefer not to say, prefer to self-describe), age, ethnicity, student status (yes/no), and whether they were diagnosed with or suspected: autism, anxiety, depression, an eating disorder, or muscle dysmorphia. Participants also reported their height (in either centimetres or feet and inches), as well as their weight (in either kilograms or stones and pounds).

Autistic traits were measured using the Autism Spectrum Quotient (AQ) [55]. The AQ is a 50-item measure that can produce a total score (ranging from 0 to 50), as well as five subscale scores (Social Skill, Attention Switching, Attention to Detail, Communication, Imagination). Each item has four response options ('definitely agree', 'slightly agree', 'slightly disagree' and 'definitely disagree') that are coded as either 0 or 1 (the direction of coding differs across items). The AQ has been used extensively to measure autistic traits in general population samples [6] and is a reliable and valid tool that can differentiate between autistic and non-autistic adults [55,56]. Internal consistency (Cronbach's $\alpha$) in the current study for the total score was $\alpha$ = 0.859; for the subscales it was as follows: Social Skill $\alpha$ = 0.770, Attention Switching $\alpha$ = 0.644, Attention to Detail $\alpha$ = 0.619, Communication $\alpha$ = 0.707, Imagination $\alpha$ = 0.557.

The Eating Attitudes Test (EAT-26) [57] was used to measure disordered eating symptoms. The EAT-26 has been used widely to measure the cognitive and behavioural symptoms of disordered eating in clinical and general population males and females [e.g., 15,58]. It is a 26-item measure which produces a total score (sum of all items) and three subscales (Dieting, Bulimia, Oral Control). Each item has six response options ranging from 0 to 3 ("always" = 3, "almost always" = 2, "often" = 1, "seldom" = 0, "hardly ever" = 0, and "never" = 0). Item 26 is reversed scored (i.e., "always", "nearly always" and "often" = 0, "seldom" = 1, "almost never" = 2, "never" = 3). In the current study, Cronbach's $\alpha$ for the EAT-26 total score was $\alpha = 0.887$, and for the subscales: Dieting $\alpha = 0.863$, Bulimia $\alpha = 0.764$, and Oral Control $\alpha = 0.651$.

The Over-Evaluation of Weight and Shape subscale from the Eating Disorder Examination Questionnaire (EDE-Q) [59] was used to measure the extent to which weight and shape influenced individuals' self-evaluation. It consists of two items: "Over the past 4 weeks, has your weight influenced how you feel about (judge) yourself as a person?" and "Over the past 4 weeks, has your body shape influenced how you feel about (judge) yourself as a person?" Participants responded on a 7-point scale ("not at all" = 1, "very slightly" = 2, "slightly" = 3, "somewhat" = 4, "moderately" = 5, "strongly" = 6, "very strongly" = 7). To analyse this subscale, a single mean score was calculated from the two items. The EDE-Q is a popular questionnaire-based assessment for disordered eating, and this subscale has been used as a standalone measure of over-evaluation of weight and shape in clinical and general population samples [60,61].

The Drive for Muscularity Scale (DMS) [26] was used to measure muscularity-oriented attitudes and behaviours. The DMS is a 15-item scale to which participants respond on a six-point scale (1 = Never to 6 = Always), with higher scores indicating higher drive for muscularity. The DMS measures drive for muscularity with a total score as well as a two-factor model [62]. The first factor *muscle-oriented body image* (7 items) measures muscle-focused body image concerns. Sample items include "I wish that I were more muscular" and "I think I would feel more confident if I had more muscle mass". The second factor *muscle-oriented behaviour* (8 items) measures muscle-focused behaviours (weight training adherence and dietary consumption in pursuit of muscularity). Sample items include "I use protein or energy supplements" and "I feel guilty if I miss a weight training session". Cronbach's $\alpha$ for DMS was as follows: total score, $\alpha = 0.920$; muscle-oriented body image, $\alpha = 0.922$; muscle-oriented behaviour, $\alpha = 0.884$.

The Hospital Anxiety and Depression Scale (HADS) [63] was used to measure symptoms of anxiety and depression. The HADS has been demonstrated to be reliable and valid in autistic and non-autistic populations [64]. It is a 14-item scale used to assess an individual's mental state over the previous two weeks. Each item is scored on a 0–3 scale (response options vary across items), with seven items summed to provide a score for anxiety, and seven items summed to provide a score for depression. Internal consistency for the HADS subscales in the current study was as follows: anxiety, $\alpha = 0.839$; depression, $\alpha = 0.764$.

## Data analysis

We pre-registered the analysis plan on the Open Science Framework (osf.io/kz6x4), and the analyses presented here only deviate from this where specified. IBM SPSS version 25 was used to analyse the data, and the effects are considered statistically significant when $p < 0.05$ (two-tailed). Effect sizes are interpreted based on Cohen [65]: small ($d = 0.20$, $r = 0.10$, $\varphi = 0.10$), medium ($d = 0.50$, $r = 0.30$, $\varphi = 0.30$), large ($d = 0.80$, $r = 0.50$, $\varphi = 0.50$).

Body mass index (BMI) was calculated as $BMI = weight (kg) / [height (m)]^2$. Independent samples $t$-tests were used to examine for sex differences in AQ total score, EAT-26 total score, DMS total score, HADS (anxiety and depression subscales), and over-evaluation of weight and

shape. Partial correlations (controlling for age and BMI) were used to examine associations between AQ and EAT-26 and between AQ and DMS. Fisher's *r*-to-*z* transformations were used to compare the slopes for correlations observed in males and females. Two multiple linear regression models were used to test whether the associations between AQ and EAT-26 and AQ and DMS were independent of concurrent anxiety and depression levels. The regression models showed no evidence of multicollinearity (all tolerance values > 0.32; all variance inflation factor (VIF) < 3.044). However, as would be expected, once both the main effects and interaction terms were included the multicollinearity statistics for the interactions were inflated and unreliable. Error was independent between predictors (Durbin-Watson ~2), data were broadly homoscedastic, and residuals were normally distributed. The first regression model included EAT-26 (total score) as the outcome, and the predictors were AQ (total score), anxiety, depression, AQ*anxiety, and AQ*depression; covariates were age, sex, BMI, and over-evaluation of weight and shape. The second regression model included DMS (total score) as the outcome, and the predictors were AQ (total score), anxiety, depression, AQ*anxiety, and AQ*depression; covariates were age, sex, BMI, and over-evaluation of weight and shape. Considering that the recruitment strategy changed during data collection to increase the male sample from Prolific, we included recruitment strategy as a predictor variable in Step 2 of the multiple regression models (note that this was not part of our original pre-registered analysis plan, but we considered it would be an appropriate adjustment considering the change to recruitment strategy). However, this variable was not a statistically significant predictor in any of the models and its inclusion did not change the overall pattern of results. We therefore removed it from the analyses presented here and continued with our original pre-registered plan.

## Results

Descriptive statistics and sex differences for the main study variables are detailed in Table 1. Females scored significantly higher on the EAT-26 and each of its subscales, whereas males scored significantly higher on the AQ, DMS, and each of the DMS subscales. Over-evaluation of weight and shape and HADS anxiety were significantly higher in females than males, but there was no sex difference for HADS depression.

**Table 1. Descriptive statistics and sex differences for the main study variables.**

|  | Females | | | Males | | | Difference | | | |
|---|---|---|---|---|---|---|---|---|---|---|
|  | *n* | *M* | *SD* | *n* | *M* | *SD* | *t* | *df* | *p* | *d* |
| AQ total score | 408 | 18.97 | 9.26 | 649 | 20.54 | 7.58 | -3.021 | 1055 | **0.003** | **0.19** |
| EAT-26 total score | 413 | 14.17 | 13.24 | 649 | 8.20 | 8.11 | 9.123 | 1060 | **<0.001** | **0.54** |
| EAT-26 dieting | 415 | 8.88 | 8.71 | 649 | 4.99 | 5.46 | 8.955 | 1062 | **<0.001** | **0.54** |
| EAT-26 bulimia and food preoccupation | 414 | 2.64 | 3.49 | 649 | 1.16 | 2.15 | 8.563 | 1061 | **<0.001** | **0.51** |
| EAT-26 oral control | 414 | 2.50 | 3.22 | 649 | 2.04 | 2.63 | 2.529 | 1061 | **0.012** | **0.16** |
| EDE-Q over-evaluation of weight and shape | 415 | 4.51 | 1.98 | 651 | 3.31 | 1.79 | 10.212 | 1064 | **<0.001** | **0.64** |
| DMS total score | 411 | 27.70 | 10.51 | 650 | 41.34 | 14.71 | -16.338 | 1059 | **<0.001** | **1.07** |
| DMS muscle-oriented body image | 413 | 15.82 | 6.85 | 650 | 24.56 | 9.11 | -16.725 | 1061 | **<0.001** | **1.08** |
| DMS muscle-oriented behaviour | 411 | 11.92 | 5.13 | 650 | 16.78 | 8.14 | -10.826 | 1059 | **<0.001** | **0.71** |
| HADS anxiety | 414 | 10.19 | 4.42 | 650 | 8.37 | 4.20 | 6.729 | 1062 | **<0.001** | **0.42** |
| HADS depression | 414 | 5.85 | 3.43 | 650 | 5.85 | 3.71 | -0.009 | 1062 | 0.993 | 0.00 |

Bold text indicates a statistically significant sex difference (*p* < 0.05). AQ = Autism Spectrum Quotient; EAT-26 = Eating Attitudes Test 26; EDE-Q = Eating Disorder Examination Questionnaire; DMS = Drive for Muscularity Scale; HADS = Hospital Anxiety and Depression Scale.

**Table 2. Partial correlations controlling for age, sex, and BMI.**

| | 1 | 2 | 3 | 4 | 5 | 6 | 7 | 8 | 9 | 10 |
|---|---|---|---|---|---|---|---|---|---|---|
| AQ total score (1) | | | | | | | | | | |
| EAT-26 total score (2) | 0.200*** | | | | | | | | | |
| EAT-26 dieting (3) | 0.148*** | 0.937*** | | | | | | | | |
| EAT-26 bulimia and food preoccupation (4) | 0.173*** | 0.805*** | 0.686*** | | | | | | | |
| EAT-26 oral control (5) | 0.151*** | 0.586*** | 0.341*** | 0.316*** | | | | | | |
| EDE-Q over-evaluation of weight and shape (6) | 0.146*** | 0.489*** | 0.537*** | 0.435*** | 0.069* | | | | | |
| DMS total score (7) | 0.077* | 0.255*** | 0.223*** | 0.223*** | 0.163*** | 0.263*** | | | | |
| DMS muscle-oriented body image (8) | 0.125*** | 0.192*** | 0.154*** | 0.172*** | 0.149*** | 0.270*** | 0.868*** | | | |
| DMS muscle-oriented behaviour (9) | -0.003 | 0.245*** | 0.230*** | 0.210*** | 0.127*** | 0.172*** | 0.832*** | 0.445*** | | |
| HADS anxiety (10) | 0.389*** | 0.296*** | 0.272*** | 0.295*** | 0.132*** | 0.326*** | 0.147*** | 0.211** | 0.028 | |
| HADS depression (11) | 0.400*** | 0.252*** | 0.207*** | 0.269*** | 0.142*** | 0.290*** | 0.103** | 0.191*** | -0.028 | 0.586*** |

*Note.* AQ Autism Spectrum Quotient, *EAT-26* Eating Attitudes Test-26, *DMS* Drive for Muscularity Scale, *HADS* Hospital Anxiety and Depression Scale.

\*\*\* $p < 0.001$

\*\* $p < 0.010$

\* $p < 0.050$.

Partial correlations (controlling for age, sex, and BMI) are presented in Table 2. All scales except for DMS muscle-oriented behaviour were significantly positively correlated with AQ total score. To examine sex differences in the correlations, we conducted partial correlation analyses in males and females separately (controlling for age and BMI). The partial correlation between AQ total score and EAT-26 total score was positive and statistically significant in both females, $r_{\text{partial}}$ (400) = 0.271, $p < 0.001$, and males, $r_{\text{partial}}$ (640) = 0.126, $p = 0.001$. A Fisher's $r$-to-$z$ transformation supported our hypothesis and replicated the finding of Barnett et al. [15], showing that the correlation between AQ score and EAT-26 score was significantly stronger in females than males, $z = 2.376$, $p = 0.009$ (Fig 1).

The partial correlation between AQ total score and DMS total score was positive and statistically significant in females $r_{\text{partial}}$ (397) = 0.122, $p = 0.015$, and positive but not statistically significant in males $r_{\text{partial}}$ (640) = 0.065, $p = 0.103$. This finding was surprising (and does not support our prediction), with the association being stronger in the female sample than the male sample. However, a Fisher's $r$-to-$z$ transformation showed this difference was not statistically significant, $z = 0.897$, $p = 0.185$ (Fig 2).

A multiple linear regression model (using the enter method) was used to test whether the association between AQ total score and EAT-26 total score was independent of anxiety and depression symptoms. A hierarchical approach was taken to determine the additional variance in the outcomes that could be predicted by HADS anxiety and HADS depression scores after taking autistic traits and the other covariates into account. The predictors/covariates were entered in the following order: Step 1: AQ total score; Step 2: sex, AQ*sex, age, BMI, over-evaluation of weight and shape; Step 3: anxiety and AQ*anxiety; Step 4: depression and AQ*depression.

The results of the first linear regression (outcome = EAT-26 total score) are presented in Table 3. Step 1 showed that AQ total score was a significant predictor of EAT-26, although it only accounted for a small amount of variance (adjusted $R^2$ = 0.029). The addition of covariates at Step 2 resulted in the model being able to account for a significantly larger proportion of the variance (adjusted $R^2$ = 0.335; adjusted $R^2$ change = 0.306, $p < 0.001$). Step 3 resulted in a small but statistically significant increase in the variance explained (Step 3: adjusted $R^2$ = 0.348; adjusted $R^2$ change = 0.013, $p < 0.001$). Finally, Step 4 resulted in a very small and not statistically significant increase in the variance explained (adjusted $R^2$ = 0.350; adjusted $R^2$

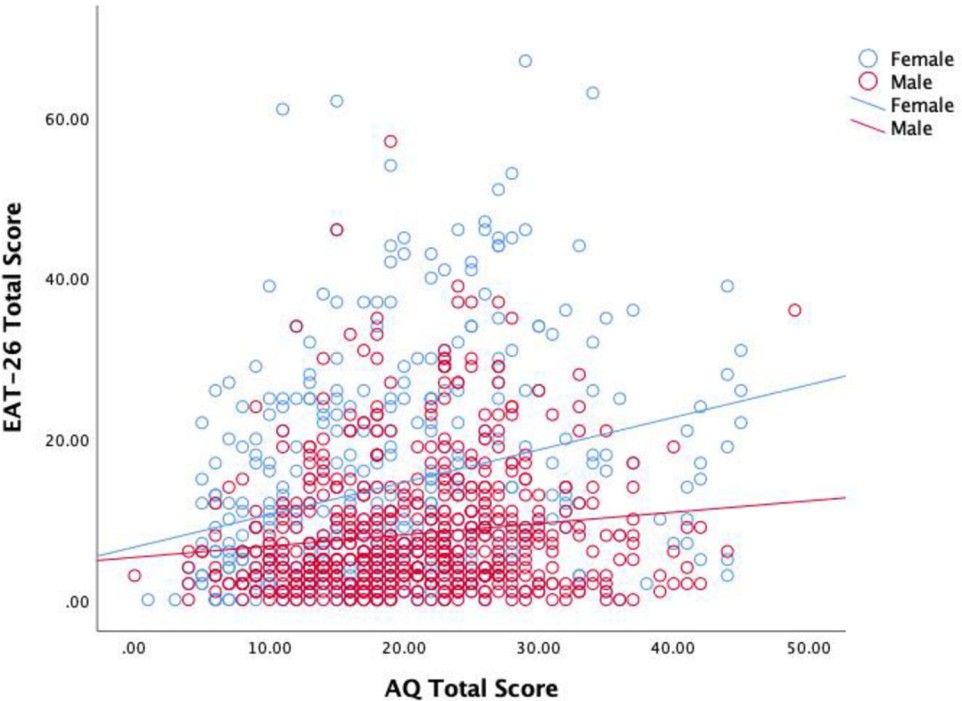

**Fig 1. Association between autistic traits and disordered eating symptoms stratified by sex.**

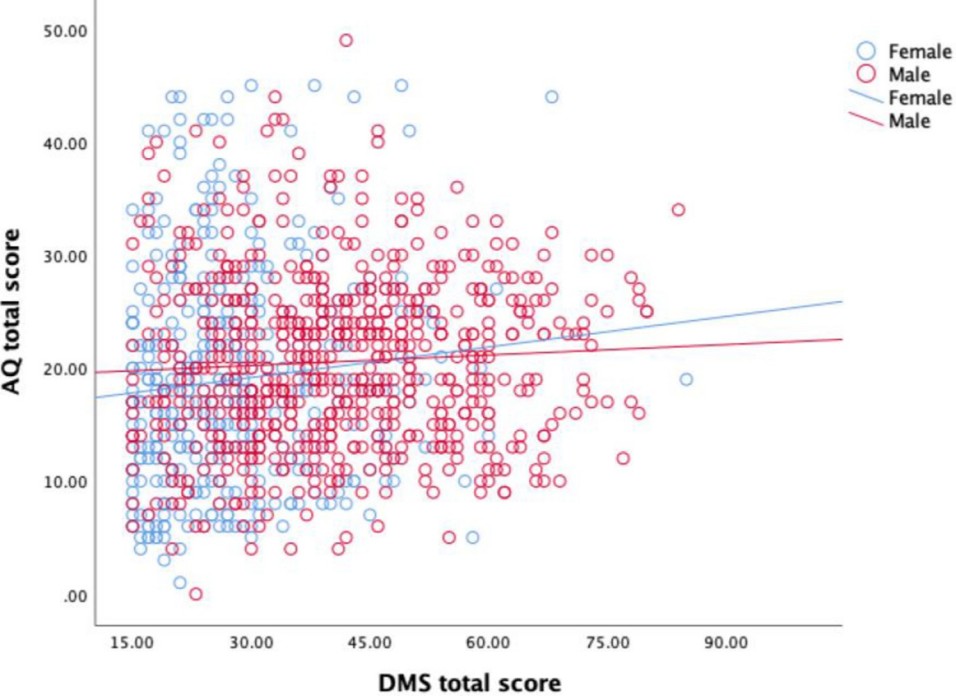

**Fig 2. Association between autistic traits and drive for muscularity stratified by sex.**

**Table 3. Hierarchical linear regression models with EAT-26 total score as the outcome.**

| | Step 1 | | | | Step 2 | | | | Step 3 | | | | Step 4 | | | |
|---|---|---|---|---|---|---|---|---|---|---|---|---|---|---|---|---|
| | Unstandardised coefficients | | Standardised coefficients | | Unstandardised coefficients | | Standardised coefficients | | Unstandardised coefficients | | Standardised coefficients | | Unstandardised coefficients | | Standardised coefficients | |
| | B | 95% CI | β | P | B | 95% CI | β | p | B | 95% CI | β | p | B | 95% CI | β | p |
| (Constant) | 10.448 | 9.806, 11.091 | | < 0.001 | 6.397 | 3.360, 9.434 | | < 0.001 | 5.406 | 2.366, 8.446 | | 0.001 | 5.755 | 2.675, 8.835 | | < 0.001 |
| AQ total score | 0.224 | 0.147, 0.302 | 0.173 | < 0.001 | 0.514 | 0.307–0.720 | 0.397 | < 0.001 | 0.393 | 0.182, 0.605 | 0.304 | < 0.001 | 0.351 | 0.134, 0.568 | 0.271 | 0.002 |
| Sex | | | | | 1.628 | -1.172, 4.429 | 0.074 | 0.254 | 1.342 | -1.454, 4.138 | 0.061 | 0.347 | 0.842 | -2.022, 3.705 | 0.038 | 0.564 |
| AQ*Sex | | | | | -0.230 | -0.359, -0.101 | -.365 | < 0.001 | -0.191 | -0.321, -0.062 | -0.304 | 0.004 | -0.170 | -0.304, -0.036 | -0.270 | 0.013 |
| Age | | | | | -0.041 | -0.088, 0.006 | -0.045 | 0.090 | -0.030 | -0.077, 0.018 | -0.033 | 0.216 | -0.029 | -0.076, 0.019 | -0.032 | 0.238 |
| BMI | | | | | 0.003 | -.0080, 0.085 | 0.002 | 0.946 | 0.015 | -0.066, 0.097 | 0.010 | 0.713 | 0.014 | -0.068, 0.095 | 0.009 | 0.745 |
| Over-evaluation | | | | | 2.661 | 2.361, 2.961 | 0.483 | < 0.001 | 2.479 | 2.169, 2.790 | 0.450 | < 0.001 | 2.446 | 2.132, 2.759 | 0.444 | < 0.001 |
| Anxiety | | | | | | | | | 0.290 | 0.147, 0.433 | 0.118 | < 0.001 | 0.243 | 0.081, 0.405 | 0.099 | 0.003 |
| AQ*anxiety | | | | | | | | | 0.020 | 0.005, 0.035 | 0.069 | 0.007 | 0.030 | 0.011, 0.048 | 0.102 | 0.002 |
| Depression | | | | | | | | | | | | | 0.127 | -0.064, 0.318 | 0.043 | 0.193 |
| AQ*Depression | | | | | | | | | | | | | -0.019 | -0.041, 0.003 | -0.054 | 0.097 |
| Model fit | $F(1, 1045) = 32.344, p < 0.001$ | | | | $F(6, 1040) = 88.912, p < 0.001$ | | | | $F(8, 1038) = 70.860, p < 0.001$ | | | | $F(10, 1036) = 57.174, p < 0.001$ | | | |
| $R^2$ | 0.030 | | | | 0.339 | | | | 0.353 | | | | 0.356 | | | |
| $\Delta R^2$ | | | | | 0.309 | | | | 0.014 | | | | 0.003 | | | |
| Adjusted $R^2$ | 0.029 | | | | 0.335 | | | | 0.348 | | | | 0.350 | | | |
| $\Delta$ Adjusted $R^2$ | | | | | 0.306 | | | | 0.013 | | | | 0.002 | | | |
| F change | | | | | $F(5, 1040) = 97.247, p < 0.001$ | | | | $F(2, 1038) = 11.379, p < 0.001$ | | | | $F(2, 1036) = 1.926, p = 0.146$ | | | |

*Note. AQ* Autism Spectrum Quotient, *BMI* Body Mass Index, *Over-evaluation*, over-evaluation of weight and shape.

change = 0.002, $p$ = 0.146). The final model (Step 4) determined five independent predictors of EAT-26 score: AQ total score, sex (female), anxiety, AQ*Anxiety, and over-evaluation of weight and shape (all in the positive direction i.e., predictive of high EAT-26 scores). The AQ*Anxiety interaction effect is outlined in the scatterplot in Fig 3. The relationship between autistic traits and disordered eating symptoms is illustrated at high ($\geq$11) and low ($<$11) HADS anxiety subscale scores. As shown in Fig 3, the relationship between autistic traits and disordered eating symptoms is stronger at high levels of anxiety than low levels of anxiety.

The results of the second linear regression (outcome = DMS total score) are reported in Table 4. Step 1 revealed that AQ score was a significant predictor of DMS score, accounting for a very small amount of variance (adjusted $R^2$ = 0.010). However, once covariates were added at Step 2, AQ score was no longer a statistically significant predictor of DMS score. The addition of the covariates at Step 2 resulted in a significant increase in predictive value (adjusted $R^2$ = 0.315; adjusted $R^2$ change = 0.305, $p < 0.001$), but this was not the case at Step 3 (adjusted $R^2$ = 0.316; adjusted $R^2$ change = 0.001, $p$ = 0.173) or Step 4 (adjusted $R^2$ = 0.316; adjusted $R^2$ change = 0.000, $p$ = 0.820). The final model determined three significant independent predictors of high DMS score: sex (male), age (younger), and over-evaluation of weight

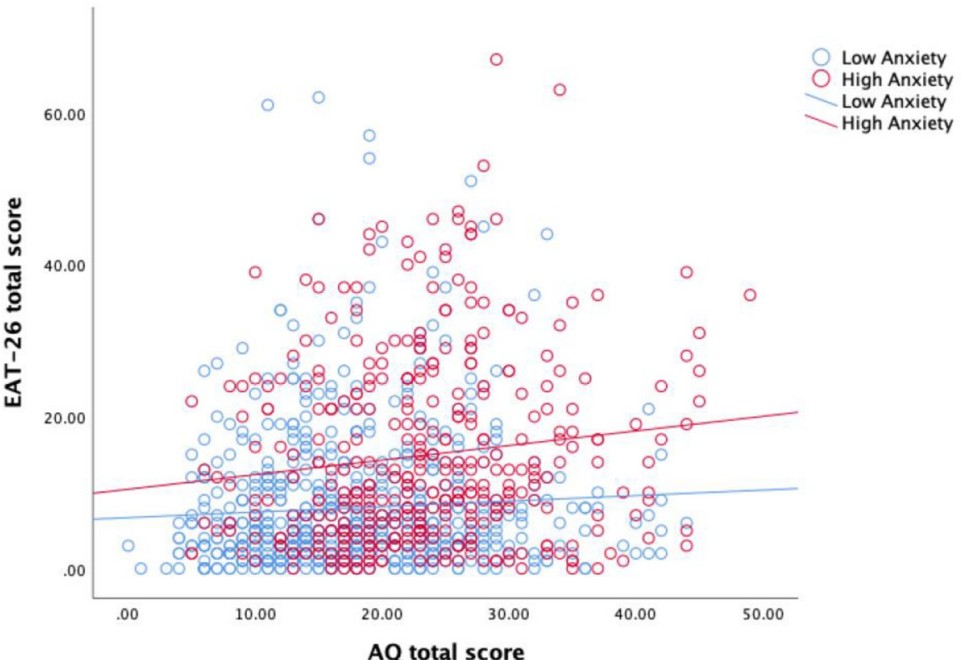

**Fig 3. Interaction between autistic traits and anxiety in the prediction of disordered eating symptoms.**

and shape (high scores). Note that low BMI ($p = 0.055$) and high anxiety ($p = 0.069$) were close to meeting the significance threshold.

## Additional exploratory (not pre-registered) analyses

Considering that the correlation between autistic traits and drive for muscularity was not statistically significant in males we conducted additional (not pre-registered) multiple linear regression analyses in males and females separately (see S1 Table). The aim here was to explore whether the significant correlation found in the female subsample was independent of anxiety and depression, as this may have been masked by the male data in the combined sample. This did indeed appear to be the case, with AQ total score significantly predicting DMS total score independently of anxiety and depression in females but not males. Considering that the correlation between autistic traits and disordered eating symptoms in the original Barnett et al. [15] paper had only been analysed with a combined (male + female) sample, we also conducted sex-stratified multiple linear regression analyses for the EAT-26 outcome. These revealed that AQ total score was positively correlated with EAT-26 total score independently of anxiety and depression symptoms in the female sample, but not in the male sample. Finally, considering that the current study observed AQ total score to correlate significantly with the DMS body image subscale ($r_{partial} = 0.125$, $p < 0.001$) but not the DMS behaviour subscale ($r_{partial} = -0.03$, $p = 0.939$), we ran sex-stratified multiple linear regression analyses with the body image subscale as the outcome variable. These revealed a significant association in females but not in males. Please refer to the supplementary materials to access the tabulated data related to these additional analyses.

## Discussion

In this study we replicated the findings of Barnett et al. [15] and extended the research to examine associations between autistic traits and drive for muscularity. Statistically significant

**Table 4. Hierarchical linear regression models with DMS total score as the outcome.**

| | Step 1 | | | | Step 2 | | | | Step 3 | | | | Step 4 | | | |
|---|---|---|---|---|---|---|---|---|---|---|---|---|---|---|---|---|
| | Unstandardised coefficients | | Standardised coefficients | | Unstandardised coefficients | | Standardised coefficients | | Unstandardised coefficients | | Standardised coefficients | | Unstandardised coefficients | | Standardised coefficients | |
| | B | 95% CI | β | P | B | 95% CI | β | p | B | 95% CI | β | p | B | 95% CI | β | p |
| (Constant) | 36.093 | 35.198, 36,989 | | < 0.001 | 11.886 | 7.627, 16.146 | | < 0.001 | 11.711 | 7.408, 16.014 | | < 0.001 | 11.525 | 7.160, 15.891 | | < 0.001 |
| AQ total score | 0.190 | 0.082, 0.298 | 0.106 | 0.001 | 0.042 | -.0246, 0.331 | 0.023 | 0.777 | 0.018 | -0.281, 0.317 | 0.010 | 0.907 | 0.035 | -0.273, 0.342 | 0.019 | 0.825 |
| Sex | | | | | 16.955 | 13.025, 20.885 | 0.556 | < 0.001 | 17.202 | 13.240, 21.164 | 0.564 | < 0.001 | 17.398 | 13.332, 21.464 | 0.571 | < 0.001 |
| AQ*Sex | | | | | 0.011 | -0.169, 0.192 | 0.013 | 0.902 | 0.007 | -0.177, 0.191 | 0.008 | 0.941 | 0.000 | -0.190, 0.189 | 0.000 | 0.997 |
| Age | | | | | -0.258 | -0.325, -0.192 | -0.208 | < 0.001 | -0.249 | -0.316, -0.182 | -0.200 | < 0.001 | -0.250 | -0.317, -0.182 | -0.201 | < 0.001 |
| BMI | | | | | -0.121 | -0.237, -0.005 | -0.056 | 0.040 | -0.114 | -0.230, 0.001 | -0.053 | 0.053 | -0.114 | -0.230, 0.002 | -0.052 | 0.055 |
| Over-evaluation | | | | | 1.823 | 1.404, 2.243 | 0.240 | < 0.001 | 1.703 | 1.263, 2.142 | 0.224 | < 0.001 | 1.720 | 1.276, 2.163 | 0.227 | < 0.001 |
| Anxiety | | | | | | | | | 0.187 | -0.015, 0.389 | 0.055 | 0.070 | 0.213 | -0.017, 0.443 | 0.063 | 0.069 |
| AQ*anxiety | | | | | | | | | -0.005 | -0.026, 0.016 | -0.012 | 0.654 | -0.008 | -0.035, 0.018 | -0.021 | 0.527 |
| Depression | | | | | | | | | | | | | -0.069 | -0.340, 0.202 | -0.017 | 0.615 |
| AQ*Depression | | | | | | | | | | | | | 0.007 | -0.024, 0.038 | 0.015 | 0.645 |
| Model fit | $F(1, 1043) = 11.899$, $p = 0.001$ | | | | $F(1, 1038) = 81.120$, $p < 0.001$ | | | | $F(1, 1036) = 61.368$, $p < 0.001$ | | | | $F(1, 1034) = 49.058$, $p < 0.001$ | | | |
| $R^2$ | 0.011 | | | | 0.319 | | | | 0.322 | | | | 0.322 | | | |
| $\Delta R^2$ | | | | | 0.308 | | | | 0.003 | | | | 0.000 | | | |
| Adjusted $R^2$ | 0.10 | | | | 0.315 | | | | 0.316 | | | | 0.316 | | | |
| $\Delta$ Adjusted $R^2$ | | | | | 0.305 | | | | 0.001 | | | | 0.000 | | | |
| F change | | | | | $F(5, 1038) = 93.904$, $p < 0.001$ | | | | $F(2, 1036) = 1.757$, $p = 0.173$ | | | | $F(2, 1034) = 0.199$, $p = 0.820$ | | | |

*Note.* AQ Autism Spectrum Quotient, *BMI* Body Mass Index, *Over-evaluation*, over-evaluation of weight and shape.

sex differences were observed for AQ (males > females), EAT-26 (females > males), DMS (males > females), over-evaluation of weight/shape (females > males), and HADS anxiety (females > males), but not HADS depression. In accordance with the findings of Barnett et al. [15], a significant positive correlation between AQ total score and EAT-26 total score was observed. As predicted, this association was significantly stronger in females than males, and the correlation remained statistically significant in the combined (male + female) sample after controlling for concurrent symptoms of anxiety and depression.

The main contribution of this study is that we found a significant positive correlation between AQ total score and DMS total score and that this was observed in the female sample but, contrary to our pre-registered hypothesis, not the male sample. Furthermore, although the correlation between AQ and DMS was not statistically significant in the combined (male + female) sample after controlling for anxiety and depression, additional (not pre-registered) regression analyses stratified by sex revealed that this effect remained statistically significant in the final model for the female sample, but not for the male sample. Our findings therefore reveal that anxious and depressive symptoms do not completely account for the overlap between autistic traits and disordered eating symptoms/drive for muscularity in females.

Equally, our findings also suggest that disordered eating symptoms and drive for muscularity in males could be explained by co-occurring anxiety and depression symptoms, rather than by autistic traits *per se*.

Considering the distinct characteristics of ideal muscularity among women [38,39], as well as its increasing centrality to female appearance ideals [40–42], investigating the specific cognitive factors associated with muscularity concerns in women and highlighting any differences with men is an important research direction. A small but growing evidence-base focusing on drive for muscularity in women indicates its clear association with body-focused [66,67] and more general [68] maladaptive psychological symptoms, so the findings of this study helpfully contribute to this emerging picture. Of particular interest is our finding that autistic traits in females are more strongly associated with both female typical (disordered eating) and male typical (drive for muscularity) outcomes. This might imply a broader underlying association between autistic traits and psychopathology in females that extends beyond typical sex differences in the phenotypic expression of eating and body image disorders. This idea is supported by research showing autism-specific mechanisms underlying disordered eating in autistic women that deviate from traditional eating disorder presentations [12].

As discussed by Barnett et al. [15], the findings of their research (and now our replication) contribute to growing arguments for clinicians to consider autistic traits in the early stages of treatment for EDs to ensure effective assessment and treatment planning [12]. This is likely to require the development of standardised treatment protocols for individuals with EDs and autistic traits, an area highlighted as a specific training need by clients and clinicians [69,70]. The present research suggests that this might also be an important consideration in BDD/MD, and that the effects observed are primarily driven by correlations observed in females. This is particularly interesting considering that females are more likely than males to camouflage or hide their autistic traits [71], that autism may generally be underdiagnosed in females [72], and that autism is sometimes misdiagnosed as anxiety or depression [73].

A strength of the current study is its large sample size, which allowed sufficient statistical power to detect small effects in stratified groups. The study also presents data on drive for muscularity in a large sample of females, an area of research which is predominantly over-populated with male-only samples [e.g., 74,75]. Additionally, we implemented Open Science practices by pre-registering our hypotheses and analysis plan. However, this study is not without limitations. Notably the cross-sectional and correlational design does not allow causal relations to be inferred. The sample was not a clinical population, and no validated screening tool was used to determine diagnostic status. Additionally, participants were not excluded from the sample based on their reported diagnostic status. Another possible limitation is that we did not measure sexual orientation, which has been identified as an important factor in predicting body weight dissatisfaction and body image concerns [76]. Thus, a different rate of heterosexual and bisexual or gay/lesbian participants in the different groups may have biased the results. Finally, as previous research has determined that autistic people, on average, have a reduced ability to accurately estimate the size of their body compared to non-autistic people [77], the present findings should be approached with caution when generalising to clinical populations. However, considering previous research has linked autism to unique body-related disorders and experiences [47,78], there is a benefit to understanding the role of autistic traits in a broader range of body-related disorders.

## Conclusions

The current study successfully replicated earlier work [15] by demonstrating that the positive correlation between autistic traits and disordered eating is stronger in females than males and

independent of concurrent levels of anxiety and depression. It also investigated a previously unconsidered factor, drive for muscularity, and its relationship with autistic traits. Our findings revealed a significant positive correlation between autistic traits and drive for muscularity in females but not males. Furthermore, the correlation in females was independent of co-occurring anxiety and depressive symptoms. The findings emphasize the moderating role of sex in disordered eating and body image disturbance, and the need to account for autistic traits when considering assessment and treatment options for individuals diagnosed with or at heightened risk of eating and body disorders.

## Supporting information

**S1 Table. Hierarchical linear regression models (not pre-registered).** Six hierarchical regression models stratified by sex with DMS total score as outcome (1a and 1b), EAT-26 as outcome (1c and 1d), and DMS body image as outcome (1e and 1f).
(DOCX)

## Acknowledgments

We would like to thank Shreyasi Das, Emma Delahunty and Kate Maher for their support with data collection.

## Author Contributions

**Conceptualization:** John Galvin, Elizabeth H. Evans, Catherine V. Talbot, Claire Wilson, Gareth Richards.

**Data curation:** John Galvin, Elizabeth H. Evans, Catherine V. Talbot, Claire Wilson, Gareth Richards.

**Formal analysis:** John Galvin, Catherine V. Talbot, Claire Wilson, Gareth Richards.

**Funding acquisition:** Elizabeth H. Evans.

**Investigation:** John Galvin, Elizabeth H. Evans, Catherine V. Talbot, Claire Wilson, Gareth Richards.

**Methodology:** John Galvin, Elizabeth H. Evans, Catherine V. Talbot, Claire Wilson, Gareth Richards.

**Project administration:** John Galvin, Elizabeth H. Evans, Catherine V. Talbot, Claire Wilson, Gareth Richards.

**Resources:** John Galvin, Elizabeth H. Evans, Catherine V. Talbot, Claire Wilson, Gareth Richards.

**Software:** John Galvin, Elizabeth H. Evans, Catherine V. Talbot, Claire Wilson, Gareth Richards.

**Supervision:** John Galvin, Elizabeth H. Evans, Catherine V. Talbot, Claire Wilson, Gareth Richards.

**Validation:** John Galvin, Elizabeth H. Evans, Catherine V. Talbot, Claire Wilson, Gareth Richards.

**Visualization:** John Galvin, Elizabeth H. Evans, Catherine V. Talbot, Claire Wilson, Gareth Richards.

**Writing – original draft:** John Galvin.

**Writing – review & editing:** John Galvin, Elizabeth H. Evans, Catherine V. Talbot, Claire Wilson, Gareth Richards.

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
