## [Decision Letter · Decision Letter 0]

1 Aug 2022

PONE-D-22-16790Associations between autistic traits and disordered eating/drive for muscularity: A replication and extension of Barnett et al. (2021).PLOS ONE

Dear Dr. Talbot,

Thank you for submitting your manuscript to PLOS ONE. After careful consideration, we feel that it has merit but does not fully meet PLOS ONE’s publication criteria as it currently stands. Therefore, we invite you to submit a revised version of the manuscript that addresses the points raised during the review process.

The reviewers had positive things to say about your study and manuscript but also had several concerns.

I will not repeat all of the points raised by the reviewers; they make their points clearly. Please consider each carefully, make revisions that you believe are appropriate.

Here are my additional recommendations:

-No details were reported regarding how data were screened for normality. Please specify.

-Did the authors check whether the statistical assumptions for regression were met before implementing the analysis? I did not see information about testing the assumptions of the model or if any correction was needed.

- Adjusted R^2^ (instead of R^2^) should be reported.

We look forward to receiving your revised manuscript.

Kind regards,

Claudio Imperatori, Ph.D

Academic Editor

PLOS ONE

Journal Requirements:

Reviewers' comments:

Reviewer's Responses to Questions

**Comments to the Author**

1. Is the manuscript technically sound, and do the data support the conclusions?

Reviewer #1: Yes

Reviewer #2: Yes

2. Has the statistical analysis been performed appropriately and rigorously? 

Reviewer #1: Yes

Reviewer #2: Yes

3. Have the authors made all data underlying the findings in their manuscript fully available?

Reviewer #1: No

Reviewer #2: Yes

4. Is the manuscript presented in an intelligible fashion and written in standard English?

Reviewer #1: Yes

Reviewer #2: Yes

5. Review Comments to the Author

Reviewer #1: The paper reports an exciting study on the relationships between AQ and ED psychopathology, looking for male/female differences. The authors have included a large sample of participants. They are also aware of the need to validate the online responses received, increasing their robustness. I have only a few comments for the authors to increase their clarity.

- in the introduction I think the authors should increase their description of connections between MD and AQ because they are not clear to me.

- please be aware of the use of gender or sex definition. In some parts of the paper, you use "sex/gender", but they are not the same. Moreover, it is not clear to me if you evaluate this aspect and how. Finally, sexual orientation is a key element for body weight dissatisfaction and body image concerns. This could be a bias in your analysis because a different rate between heterosexual and bisexual or gay/lesbian participants could unbalance the results (see https://doi.org/10.1007/s40519-020-01047-7).

- body image concerns seem to be linked to cognitive evaluation and judgments (see https://doi.org/10.1002/erv.2812). I think this aspect has a role in your study due to the connections between AQ traits and executive functions and the ability to interpret judgments. Please, consider this aspect in your discussion.

- in the discussion, the authors have poorly addressed the differences between males and females. I think this is an interesting aspect that needs more space, especially for the discussed literature about the presence of autistic traits in males and ED psychopathology in women.

- While methods are clear and pre-registered, I think they should use more conservative p-values due to the presence of a very large number of analyses. Their p-values are reported al <.001 but with a large number of comparisons, a corrected p-value could be also lower. Please consider this aspect.

Reviewer #2: Dear Editor, first of all thank you for giving me the opportunity to review this interesting proposal.

The manuscript “Associations between autistic traits and disordered eating/drive for muscularity: A replication and extension of Barnett et al. (2021)” is well organized and well written. The paper shed some like on the link between autism and eating symptoms, in particular drive for muscularity. It has the strength of emphasizing differences in sex among.

I suggest some minor revision:

1) For esthetic sake, I suggest that the title do not include the reference with the data. I suggest rephrasing the title.

2) The abstract needs to be rewritten, avoiding the numbered list, controlling spaces and phrases. Furthermore the last sentence: “The findings have both research and clinical implications” is too vague and need to be specified in some way.

3) Results consider anxiety and depression as confounding factors but in the introductions those two conditions are not addressed properly. I suggest to add a paragraph and some references as a rationale for the analysis.

4) Among the limitations of the paper should be described the fact that the sample is not a clinical population and that no diagnosis by experienced clinician has been conducted. Authors choose not to have exclusion criteria like having an actual diagnosis of ed. This should be addressed in the limitation.

6. PLOS authors have the option to publish the peer review history of their article (what does this mean?). If published, this will include your full peer review and any attached files.

Reviewer #1: No

Reviewer #2: No

---

## [Author Response · Author response to Decision Letter 0]

14 Sep 2022

Dear Reviewers,

Thank you very much for taking the time to review our paper and providing these excellent comments to help us improve the manuscript. The comments are fair and constructive, and we appreciate your input.

Responses to each comment are provided below. The changes in the manuscript are highlighted with red text.

All the best,

Corresponding Author

Editor

• In response to the comments by the Editor we have now included information regarding the statistical assumptions for the regression models in the data analysis section of the method. We have also amended the results to include the numbers for adjusted R2.

Reviewer #1

The paper reports an exciting study on the relationships between AQ and ED psychopathology, looking for male/female differences. The authors have included a large sample of participants. They are also aware of the need to validate the online responses received, increasing their robustness. I have only a few comments for the authors to increase their clarity.

• Thank you for reviewing the work and for these valuable comments.

- in the introduction I think the authors should increase their description of connections between MD and AQ because they are not clear to me.

• We have now included a new section in the introduction labelled “rationale for the current investigation” and within this section we have increased our description of connections between MD and AQ (please see end of page 6 + page 7).

- please be aware of the use of gender or sex definition. In some parts of the paper, you use "sex/gender", but they are not the same. Moreover, it is not clear to me if you evaluate this aspect and how. Finally, sexual orientation is a key element for body weight dissatisfaction and body image concerns. This could be a bias in your analysis because a different rate between heterosexual and bisexual or gay/lesbian participants could unbalance the results (see https://doi.org/10.1007/s40519-020-01047-7).

• In the questionnaire we specifically asked the participants to report their sex. We have therefore changed the terminology in the paper to reflect this. We did not measure sexual orientation and therefore have included this point in the limitations section of the discussion and cited the suggested paper. 

- body image concerns seem to be linked to cognitive evaluation and judgments (see https://doi.org/10.1002/erv.2812). I think this aspect has a role in your study due to the connections between AQ traits and executive functions and the ability to interpret judgments. Please, consider this aspect in your discussion.

• As we do not measure executive functions or judgements in this study this aspect does not form part of our discussion. We feel it would take the findings of the study too far to speculate on this aspect considering we only have correlational data on the global measures of AQ, ED and DMS. However, we agree this is relevant background information regarding the rationale for focusing on autistic traits and have therefore included it as a point in the introduction section (paragraph 7) and cited the suggested paper.

- in the discussion, the authors have poorly addressed the differences between males and females. I think this is an interesting aspect that needs more space, especially for the discussed literature about the presence of autistic traits in males and ED psychopathology in women.

• We have now added some additional literature/information in this regard in the discussion. 

- While methods are clear and pre-registered, I think they should use more conservative p-values due to the presence of a very large number of analyses. Their p-values are reported al <.001 but with a large number of comparisons, a corrected p-value could be also lower. Please consider this aspect.

• We understand this concern but feel that in this case correcting for the number of tests may confuse the interpretation of results and increase the risk of a type II error. Although for completeness the tables in the paper report the statistics for all the variables (including subscale scores), and we have also conducted some additional exploratory analyses, please note that the main hypotheses as per our pre-registration plan involved only n=9 tests. To be more specific: tests 1-3 compared males and females on AQ score, EAT-26 score, and DMS score; tests 4 and 5 involved partial correlations between AQ score and EAT-26 score and AQ score and DMS score; tests 6 and 7 were Fisher r-to-z tests to examine sex differences between males and females on the aforementioned correlations; and finally, tests 8 and 9 included the two multiple regression analyses (one with EAT-26 score and the other DMS score as the outcome). 

Reviewer #2

Dear Editor, first of all thank you for giving me the opportunity to review this interesting proposal.

The manuscript “Associations between autistic traits and disordered eating/drive for muscularity: A replication and extension of Barnett et al. (2021)” is well organized and well written. The paper shed some like on the link between autism and eating symptoms, in particular drive for muscularity. It has the strength of emphasizing differences in sex among.

• Thank you for reviewing the work and for your valuable comments.

I suggest some minor revision:

1) For esthetic sake, I suggest that the title do not include the reference with the data. I suggest rephrasing the title.

• We have now rephrased the title and removed the reference. It now reads: The associations between autistic traits and disordered eating/drive for muscularity are independent of anxiety and depression in females but not males.

2) The abstract needs to be rewritten, avoiding the numbered list, controlling spaces and phrases. Furthermore the last sentence: “The findings have both research and clinical implications” is too vague and need to be specified in some way.

• We have removed the numbered list, re-worded some of the text, and provided more specificity as suggested.

3) Results consider anxiety and depression as confounding factors but in the introductions those two conditions are not addressed properly. I suggest to add a paragraph and some references as a rationale for the analysis.

• We have added a paragraph in the introduction that explains the rationale for including anxiety and depression as confounding factors (see paragraph 8 of the introduction).

4) Among the limitations of the paper should be described the fact that the sample is not a clinical population and that no diagnosis by experienced clinician has been conducted. Authors choose not to have exclusion criteria like having an actual diagnosis of ed. This should be addressed in the limitation.

• We have now added these points into the limitations section.

---

## [Decision Letter · Decision Letter 1]

4 Oct 2022

The associations between autistic traits and disordered eating/drive for muscularity are independent of anxiety and depression in females but not males

PONE-D-22-16790R1

Dear Dr. Talbot,

We’re pleased to inform you that your manuscript has been judged scientifically suitable for publication and will be formally accepted for publication once it meets all outstanding technical requirements.

Kind regards,

Claudio Imperatori, Ph.D

Academic Editor

PLOS ONE

Additional Editor Comments (optional):

Reviewers' comments:

Reviewer's Responses to Questions

**Comments to the Author**

1. If the authors have adequately addressed your comments raised in a previous round of review and you feel that this manuscript is now acceptable for publication, you may indicate that here to bypass the “Comments to the Author” section, enter your conflict of interest statement in the “Confidential to Editor” section, and submit your "Accept" recommendation.

Reviewer #1: All comments have been addressed

Reviewer #2: All comments have been addressed

2. Is the manuscript technically sound, and do the data support the conclusions?

Reviewer #1: Yes

Reviewer #2: Yes

3. Has the statistical analysis been performed appropriately and rigorously? 

Reviewer #1: Yes

Reviewer #2: Yes

4. Have the authors made all data underlying the findings in their manuscript fully available?

Reviewer #1: No

Reviewer #2: Yes

5. Is the manuscript presented in an intelligible fashion and written in standard English?

Reviewer #1: Yes

Reviewer #2: Yes

6. Review Comments to the Author

Reviewer #1: I think the authors have addressed all my comments. The paper is really improved with the revision. Great job!

Reviewer #2: (No Response)

7. PLOS authors have the option to publish the peer review history of their article (what does this mean?). If published, this will include your full peer review and any attached files.

Reviewer #1: No

Reviewer #2: **Yes: **Matteo Panero

---

## [Editor Report · Acceptance letter]

6 Oct 2022

PONE-D-22-16790R1 

The associations between autistic traits and disordered eating/drive for muscularity are independent of anxiety and depression in females but not males 

Dear Dr. Talbot:

I'm pleased to inform you that your manuscript has been deemed suitable for publication in PLOS ONE. Congratulations! Your manuscript is now with our production department. 

Kind regards, 

on behalf of

Dr. Claudio Imperatori 

Academic Editor

PLOS ONE